# Beyond medical prescription: Unveiling coping strategies' role between perceived quality of care and treatment adherence among hypertensive patients

Prince Owusu Adoma[1]*, Francis Aacquah[1], Christopher Bawiah[2], Samuel Oke[3], Abubakari Yakubu[4]

1 Department of Health Administration and Education, University of Education Winneba, Winneba, Central Region, Ghana, 2 Department of Health Information, College of Health, Yamfo, Bono East Region, Ghana, 3 Department of Public Health, Catholic University of Ghana, Sunyani, Bono Region, Ghana, 4 Business and Resource Centre-Yendi, Rural Enterprise Programme, Ministry of Trade, Accra, Ghana

* poadoma@uew.edu.gh

## Abstract

Beyond medical prescription for hypertension treatment, management requires consistent level of adherence, high quality of care and adaptive coping strategies, which are rarely considered. Our study analysed the moderating role of coping strategies in the relationship between perceived quality of care and treatment adherence among hypertensive patients. A hospital-based retrospective observational cohort study was conducted among adult hypertensive patients who attended clinic at a tertiary hospital between August 2023 and July 2024. A total of 621 consenting patients were purposively surveyed using three pre-existing instruments: Adherence to Systemic Hypertension Treatment Scale, the Service Quality Questionnaire, and the Coping Inventory for Stressful Situations-21 (CISS-21). Data were analysed using correlation and bootstrap moderation analyses. The correlation matrix showed weak positive associations between age and adherence (Spearman's rho = 0.098, p = 0.014), comorbidities and adherence (Spearman's rho = 0.068, p = 0.091), as well as BP and comorbidities (Spearman's rho = 0.074, p = 0.065). Males were more likely to exhibit lower adherence to hypertension treatment compared to females. As females' perception of quality care increased, so did their adherence to treatment. Among the three coping strategies, only task-oriented coping significantly moderated the relationship between perceived quality of care and treatment adherence (B = 0.268, 95% CI = 0.17–0.365). While sex differences should be considered in treatment adherence initiatives, task-oriented coping may play a crucial role in improving adherence, particularly among individuals with a positive perception of care quality. Addressing sex-specific barriers and promoting task-oriented coping strategies could enhance treatment adherence.

**Data availability statement:** All data can be found in the manuscript and supporting information files.

**Funding:** The authors received no specific funding for this work.

**Competing interests:** The authors have declared that no competing interests exist.

## Introduction

Hypertension management remains a challenge, as treatment focuses primarily on antihypertensive therapy (AHT) for effective blood pressure (BP) control [1]. Beyond the prescription for AHT, hypertension management requires consistent level of adherence to treatment, perception of high quality of care and adaptive coping strategies, which are behavioural in nature. However, more often than not, these practices are rarely measured in the healthcare system. According to Bhandari et al., [2], the practices can be categorised into themes; misconceptions about hypertension and its treatment, behavioural modification difficulties (capability barriers), faith in alternative medicine and fear of treatment consequences (motivation barriers), as well as poor patient-provider communication, stigma, and socio-cultural influences (opportunity barriers). These practices are uncovered and have generally undergone fewer and less rigorous research, creating unclear an relationship between behavioural interventions and hypertension control. This makes it difficult to quantify the main contribution of behaviour interventions on hypertension control.

In the healthcare system, cognitive and behavioural factors such as lifestyle, coping strategies, treatment adherence, and perceptions of quality of care often receive less attention than medical and pharmacological interventions. Evidence suggests that it is inevitable to control hypertension without adopting a healthy lifestyle, adaptive coping, adherence to treatment and positive perception on quality of care [3–5]. These patient-related factors play a crucial role in hypertension control and have significantly contributed to the challenges in achieving BP targets [6].

Despite the widespread use of antihypertensive medication (AHM) for hypertension treatment [7], a significant gap remains in achieving optimal blood pressure control [8]. This challenge may stem from a generalised approach to hypertension management, where behavioural interventions are often overlooked or inadequately monitored. As a result, AHM alone is insufficient to effectively control high BP and meet treatment targets

The global target for non-communicable diseases (NCDs) is to reduce the prevalence of hypertension by 33% between 2010 and 2030 [9]. One effective approach to achieving this goal is personalised care that addresses the unique needs of each patient. In line with this, personalising care has become central to hypertension treatment, as patients who actively engage in shared decision-making are more likely to experience better health outcomes [9,10].

Evidence suggests that hypertension management is associated with increased stress, which negatively impacts treatment targets depending on the coping mechanisms used. [5,11,12]. Thus, coping strategies are essential in hypertension management, as they can either positively or negatively influence the relationship between the perception of quality of care and adherence to treatment, ultimately affecting BP control and overall health outcomes.

The perception of quality of care plays a critical role, as patients' perceptions of care quality motivate their participation and adherence to treatment recommendations [13,14]. Adherence to treatment is essential for achieving treatment targets and reducing the risk of complications such as cardiovascular diseases and stroke

[15,16]. Understanding whether coping strategies moderate the relationship between quality of care and treatment adherence is crucial, as this behavioural insight could inform targeted interventions that improve patient outcomes and support more effective hypertension management.

## Theoretical consideration

The Individual and Family Self-Management Theory (IFSMT) [17] underpins this study, emphasising the crucial role of both individuals and families in managing chronic health conditions such as hypertension. The theory posits that both personal and external factors, including individual knowledge, skills, decision-making, social support, and healthcare provider involvement influence self-management [18]. Additionally, IFSMT highlights that self-management is a dynamic process shaped by contextual factors that can either facilitate or hinder an individual's ability to manage their health effectively.

Within this framework, coping strategies and perceptions of care quality may be presented as contextual factors that can function as both risks and protective elements that directly affect self-management efforts. As described by Lazarus & Folkman (1984), coping strategies refer to how patients manage the stress associated with chronic conditions, ultimately affecting their ability to adhere to treatment regimens. Similarly, research by Hogg, (2022) [13] and Tan et al., (2017) [14] highlights that patients' perceptions of care quality significantly shape their willingness to engage with and adhere to treatment plans.

## Purpose of the study

This study specifically analysed the moderating role of coping strategies in the relationship between perceived quality of care and treatment adherence. By doing so, we investigate how contextual factors such as coping strategies and perceptions of quality of care interact in ways that either support or hinder self-management, as outlined in IFSMT. Based on these theoretical foundations, this study explores the following hypotheses:

**H1:** There is a significant relationship between treatment adherence and health outcomes such as blood pressure (BP) control, hospital admissions, and comorbidities.

**H2:** The perception of quality of care influences hypertension treatment adherence.

**H3:** Coping strategies moderate the effect of the perception of quality of care on hypertension treatment adherence.

By investigating the moderating role of coping strategies (ref **Fig 1**), this research seeks to fill the gap in understanding how these strategies impact the relationship between perceived quality of care and treatment adherence. This study is particularly significant for developing countries like Ghana, where healthcare resources are limited, the burden of hypertension is high [19], and hypertension care is typically generalised rather than personalised. Focusing on personalised self-care strategies that are effective for BP control may improve hypertension management in such contexts.

## Materials and methods

### Study design and setting

A hospital-based retrospective observational cohort study of adult hypertensive patients who attended a clinic at a tertiary hospital in Ghana was carried out from 7th August 2023–26th July 2024. The hospital is a state-owned facility, run by the government of Ghana and supervised by the Ministry of Health (MoH), Ghana. This is a tertiary care facility with a catchment area that includes Sunyani, the capital of the Bono region, and surrounding areas. On average, 150–200 outpatients with various cardiovascular diseases attend the Diabetic and Hypertensive Clinic. Since it is a state-owned and not-for-profit health facility, accredited by the National Health Insurance Scheme, it offers free and low-cost services, attracting a larger number of patients. The facility was chosen due to its strategic location, to increase the reliability of study results and ensure generalisation of the data to the Ghanaian people.

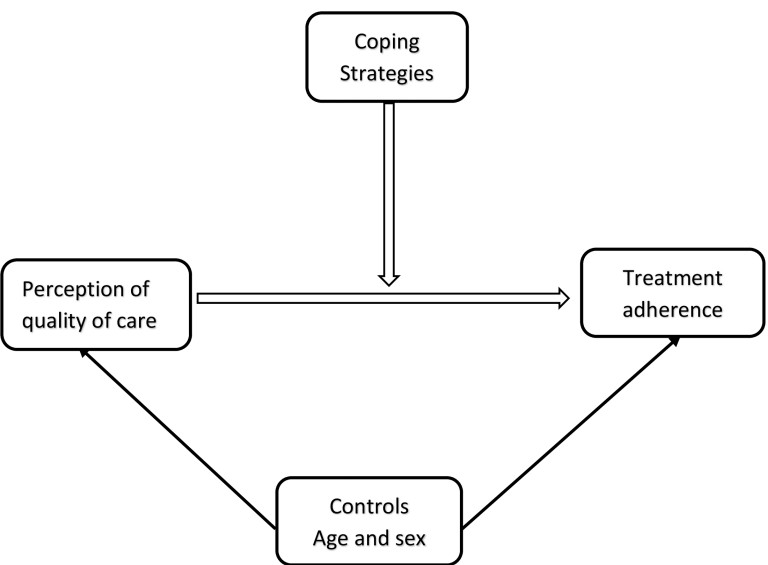

**Fig 1. An interplay of coping strategies, perceptions and treatment adherence among hypertensive patients.**

## Population

Adult hypertensive patients who attended a clinic at a tertiary hospital, Sunyani Teaching Hospital, in Ghana between August 2023 and July 2024 were targeted for the study.

**Inclusion and exclusion criteria.** The population included all patients in the cohort recruited for the purposes of this study. They included all adult hypertensive patients, diagnosed with hypertension by a professional medical practitioner, and consistently attending clinic for hypertension control at Sunyani Teaching Hospital. Additionally, only patients who provided their consent to participate in the study were recruited. Specifically, patients aged 35 years and above, who have sought hypertension care for at least six months, and had previously attended clinic at the Sunyani Teaching Hospital (STH) qualified to be involved in the study. However, newly diagnosed patients with hypertension at first visit and patients with a critical or cognitive impairment were excluded. In addition, pregnancy-induced hypertension, contraceptive pills/injectables or hormone replacement-induced hypertension, as well as those patients who were hard to trace were excluded.

## Sampling and sample size

A total of 621 adult hypertensive patients were enrolled into the study. Although all the hypertensive patients (717) in the cohort attending the clinic at STH were targeted, some of them in the cohort died (18), others dropped out of treatment (43), some could not be traced (28), and the rest refused consent (7). A consecutive sampling was used in recruiting all the patients into the study based on their availability until the total sample was reached.

## Development and validation of questionnaire

The questionnaire was developed from three different pre-existing instruments: Adherence to Systemic Hypertension (QASH) Treatment [19] to measure treatment adherence; Service Quality (SERVQUAL) Questionnaire to determine quality of service, and Coping Inventory for Stressful Situation-21 (CISS-21) Questionnaire [20] to measure coping strategies used for hypertension management.

The final instrument contained 24 items, in 4 sections (A, B, C and D). Section A contained socio-demographic variables, B on adherence to treatment, C measured perception on quality of care and finally, D on coping strategies. Patients

who responded to the items on a five-point scale, from strongly disagree to strongly agree (1–5), with high scores indicating higher use of behaviour.

The instrument was edited and tested to ensure validity and reliability. Also, 2 public health consultants from a local university assessed the instruments. The instrument was assessed by 6 adult hypertensive patients to check its clarity. Afterwards, some items were rephrased to better their meaning. The questionnaire was later pre-tested with sample data of 124 adult hypertensives. The data produced Cronbach's Alpha reliability coefficient scores of 0.80, 93.2 and 79.8 for adherence, perception on quality of care and coping strategies, respectively. Relevant covariates such as age, blood pressure (measured as the average of diastolic and systolic values), comorbidity, and admission rate were measured and treated as continuous variables.

## Data collection procedure

The cohort study spanned from August 2023 to July 2024; however, data collection for adult hypertensive patients was enumerated between 2nd September and 29th November 2024 at the Hypertensive Clinic at STH. Two research assistants (RAs) were recruited and trained for data collection. The RAs were degree holders who had no relationship with the patients. The training focused on questionnaire administration and interpretation of items from English to Twi (local) language for patients who could not read and understand English. Data collection lasted for about three months since the study aimed at reaching all participants in the cohort seeking hypertension care. In all, 621 patients were obtained and made to respond to all items in the questionnaire as they waited to see their doctors. Patients were allowed to fill in the questionnaire and return to the hospital, while others' contact details were taken to follow up for the questionnaire later at their convenience.

The major problem encountered during data collection was caregivers' inability to wait for patients to respond to all items; some patients also felt too tired and weak, in the process unable to answer all items in the questionnaire, and others were hard to trace for data collection. We also provided water and beverage drinks for a few of the patients during data collection who requested that. All patients either gave verbal or written consent before taking part in the study. For the verbal consent, participants were asked categorically if they were comfortable and ready to respond to the items in the questionnaire. Those who expressly stated "yes, I am ready" were made to answer the questionnaire.

## Statistical analysis

Statistical analysis began by describing the demographic characteristics of the study sample using measures of central tendency such as mean. This was followed by a spearman's correlation analysis to examine the relationships between adherence and health outcomes and a simple linear regression exploration of gender differences in treatment perception and adherence using scatter plots. Due to the presence of three moderators and the choice of SPSS software, Hayes' Process was not employed for the moderation analysis. Instead, moderation was assessed by including interaction terms in a multiple linear regression model. Additionally, a bootstrap technique was applied to relax parametric assumptions and provide more robust and reliable estimates [21,22]. The moderation analysis was performed in three steps using SPSS version 29: first, calculating the means for the moderating variables (task-oriented coping, avoidance-oriented coping, and emotion-oriented coping) and the independent variable (perceived quality of care); second, mean centering both the moderating and independent variables; and third, creating interaction terms between each moderating variable and the independent variable.

## Ethics statement

Ethical clearance was given by the Ghana Health Service Ethics Review Committee (ID: GHS-ERC: 041/10/23).

## Results

### Demographic characteristics of respondents

As shown in Table 1, the participants' ages ranged from 32 to 96 years, with a mean age of 59.5 years. The religious composition was predominantly Christian (78.2%), followed by Islamic (10.5%), Traditional (9.5%), and Other (1.7%) belief systems. Regarding educational background, 24.6% had no formal education, 53.8% had basic education, and 21.6% had post-basic education.

### Treatment adherence and health outcomes

As shown in **Table 2**, the correlations among the variables were generally weak. A weak positive correlation between age and adherence was observed (Spearman's rho = 0.098, p = 0.014), indicating that older individuals tended to show slightly higher adherence. However, no significant correlations were found between age and blood pressure (BP), age and comorbidities, or age and admission rate.

Adherence also showed weak correlations with health outcomes. It had a very weak, non-significant positive correlation with BP (Spearman's rho = 0.02, p = 0.626), and a weak, marginally significant positive correlation with comorbidities (Spearman's rho = 0.068, p = 0.091). BP demonstrated a weak, positive correlation with comorbidities (Spearman's rho = 0.074, p = 0.065). Interestingly, no significant relationship was found between comorbidities and admission rate, which was an unexpected finding.

### Gender differences in perception of quality of care and treatment adherence

Fig 2 illustrates the gender differences in treatment compliance and perception of quality care. The Fig 2 shows that, as males' perception of quality care increases, their adherence to treatment remains lower compared to females. In contrast, for females, higher perceptions of quality care are associated with increased treatment adherence.

**Table 1. Demographic characteristics of respondents.**

| Variables | Values | N | % | MIN | MEAN | MAX | MODE |
|---|---|---|---|---|---|---|---|
| Sex assigned at birth | Female | 427 | 68.8 | | | | |
| | Male | 194 | 31.2 | | | | |
| | Total | 621 | 100 | | | | |
| Age | | | | 32 | 59.5 | 32 | 96 |
| Marriage status | marriage | 433 | 69.7 | | | | |
| | Single | 188 | 30.3 | | | | |
| | Total | 621 | 100 | | | | |
| Religion | Christian | 486 | 78.2 | | | | |
| | Islamic | 65 | 10.5 | | | | |
| | Traditional | 59 | 9.5 | | | | |
| | Others | 11 | 1.7 | | | | |
| | Total | 621 | 100 | | | | |
| Education | No education | 153 | 24.6 | | | | |
| | Basic | 334 | 53.8 | | | | |
| | post-basic | 134 | 24.6 | | | | |
| | Total | 621 | 100 | | | | |

**Table 2. Correlation analysis of age, adherence and health outcomes.**

| | | Age | Adherence | BP | Comorbidities | Admission Rate |
|---|---|---|---|---|---|---|
| **Age** | Spearman's rho | — | | | | |
| | df | — | | | | |
| | p-value | — | | | | |
| **Adherence** | Spearman's rho | 0.098* | — | | | |
| | df | 619 | — | | | |
| | p-value | *0.014* | — | | | |
| **BP** | Spearman's rho | -0.004 | 0.02 | — | | |
| | df | 619 | 619 | — | | |
| | p-value | 0.918 | 0.626 | — | | |
| **Comorbidities** | Spearman's rho | -0.02 | 0.068 | 0.074 | — | |
| | df | 619 | 619 | 619 | — | |
| | p-value | 0.622 | *0.091* | *0.065* | — | |
| **Admission Rate** | Spearman's rho | -0.049 | -0.068 | 0.025 | 0.007 | — |
| | df | 618 | 618 | 618 | 618 | — |
| | p-value | 0.224 | *0.09* | 0.53 | 0.87 | — |

Highlighted = significant at 90% confidence interval.

Scatterplot

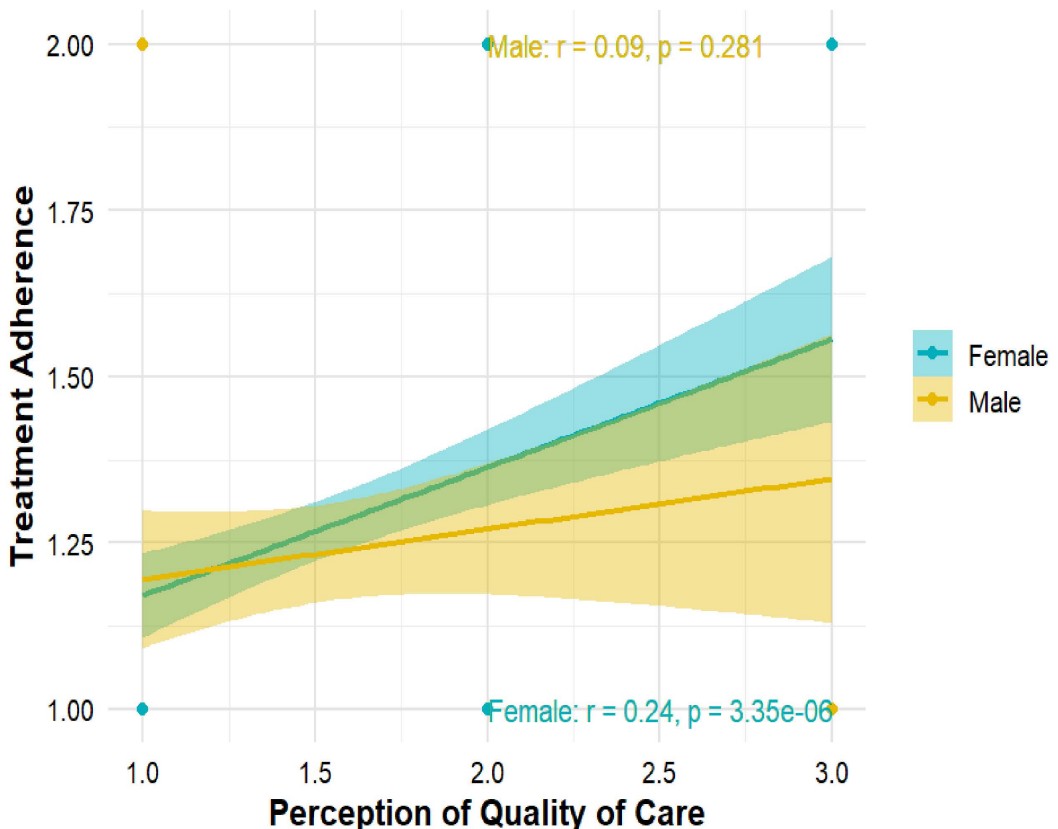

**Fig 2. Gender differences in perception of quality of care and treatment adherence.**

## Coping strategies moderating perception of quality of care and treatment adherence

In **Table 3**, the model fitness of two regression models is presented. Model 1 includes only age and sex as predictors, while Model 2 incorporates set of predictors, including age, sex, Perception of Quality, Avoidance-oriented coping, Task-oriented coping, Emotional-oriented coping, and interaction terms between these coping strategies and the Perception of Quality variable. Model 2 demonstrates significantly improved model fitness compared to Model 1. The R² value of 0.199 indicates that approximately 19.9% of the variance in hypertension treatment adherence can be explained by the predictors in this model. Additionally, the adjusted R² value of 0.187 adjusts for the number of predictors in the model, suggesting that 18.7% of the variance is explained while accounting for model complexity. The F-statistic of 20.426 is statistically significant at $p < 0.001$, indicating that the overall model is a good fit for the data. This suggests that the inclusion of Perception of Quality, coping strategies, and their interaction terms significantly enhances our ability to predict hypertension treatment adherence beyond the effects of age and sex alone. Collinearity diagnostics (S1 Text and S2 Text) can be found in the supplementary file.

**Table 4** displays the bootstrap results for the coefficients in both **Model 1** and **Model 2**. In Model 1, which comprises Age and Sex as predictors, the bootstrap analysis indicates that both Age (β = 0.002, p = 0.069) and Sex (β = 0.067,

**Table 3. Model fitness.**

| Model | R | R² | Adjusted R² |  | R² change | F change | df1 | df2 | Sig. F change |
|---|---|---|---|---|---|---|---|---|---|
| 1 | .106a | 0.011 | 0.008 | 0.412 | 0.011 | 3.524 | 2 | 618 | 0.03 |
| 2 | .445b | 0.199 | 0.187 | 0.187 | 0.187 | 20.426 | 7 | 611 | <.001 |

aPredictors: (Constant), Age, Sex.

bPredictors: (Constant), Age, Sex, Perception of Quality, Avoidance-oriented, Task-oriented, Emotional-oriented, INTCEPQ (interaction term between Emotional-oriented and Perception of Quality), INTCAPQ (interaction term between Avoidance-oriented and Perception of Quality and INTCTPQ (interaction term between Task-oriented and Perception of Quality).

**Table 4. Bootstrap for model coefficients.**

|  |  | B | Bias | Std. Error | Sig. (2-tailed) | 95% Confidence Interval | |
|---|---|---|---|---|---|---|---|
|  |  |  |  |  |  | Lower | Upper |
| 1 | (Constant) | 0.967 | 0 | 0.099 | <.001 | 0.769 | 1.163 |
|  | Sex | 0.067 | -6.93E-05 | 0.034 | 0.046 | 0.0003 | 0.132 |
|  | Age | 0.002 | 1.17E-05 | 0.001 | 0.069 | 0.00 | 0.005 |
| 2 | (Constant) | 0.639 | -0.001 | 0.121 | <.001 | 0.397 | 0.881 |
|  | Sex | 0.006 | -0.001 | 0.033 | 0.875 | -0.059 | 0.069 |
|  | Age | -0.001 | 4.44E-06 | 0.001 | 0.464 | -0.003 | 0.002 |
|  | Perception of Quality | 0.092 | 0 | 0.028 | **<.001** | 0.036 | 0.146 |
|  | Avoidance-oriented | -0.015 | 9.39E-05 | 0.024 | 0.524 | -0.062 | 0.032 |
|  | Task-oriented | 0.332 | 0.001 | 0.034 | **<.001** | 0.267 | 0.399 |
|  | Emotional-oriented | -0.014 | -4.90E-05 | 0.022 | 0.519 | -0.058 | 0.028 |
|  | INTCEPQ | 0.005 | 0.001 | 0.041 | 0.897 | -0.074 | 0.088 |
|  | INTCAPQ | -0.064 | -0.001 | 0.047 | 0.171 | -0.156 | 0.029 |
|  | INTCTPQ | 0.268 | 0.001 | 0.05 | **<.001** | 0.17 | 0.365 |

Bootstrap results are based on 5000 bootstrap samples.

1. Predictors: (Constant), Age, Sex.

2. Predictors: (Constant), Age, Sex, Perception of Quality of care, Avoidance-oriented, Task-oriented, Emotional-oriented, INTCEPQ (interaction term between Emotional-oriented and Perception of Quality), INTCAPQ (interaction term between Avoidance-oriented and Perception of Quality and INTCTPQ (interaction term between Task-oriented and Perception of Quality).

p = 0.046) exhibit statistically significant effects, albeit to a certain extent, on hypertension treatment adherence. These effects become particularly evident when considering a 90% confidence interval. In Model 2, which incorporates a broader set of predictors and interaction terms, several variables are found to have statistically significant effects on hypertension treatment adherence. Specifically, Perception of Quality (β = 0.092, p < .001), Task-oriented coping (β = 0.332, p < .001), and INTCTPQ (β = 0.268, p < .001) have positive effects, while avoidance-oriented coping does not significantly contribute to explaining variation in treatment adherence, showing a non-significant effect. Additionally, the interaction terms INTCEPQ and INTCAPQ also exhibit non-significant effects. The study's results indicate that among the three coping strategies considered, only task-oriented coping serves as a moderator in the relationship between the perception of quality care and treatment adherence.

The study proceeded with a simple slope analysis to understand the moderating impact of task-oriented coping on the relationship between perception of quality of care and treatment compliance. The outcomes presented in Table 5 reveal that the relationship between Perception of Quality of care and treatment compliance is statistically significant and positive when Task-oriented coping is at an average or high level but insignificant at a low level. This finding suggests that the effect of perception of quality of care on treatment compliance varies depending on the level of Task-oriented coping. Fig 2 provides a graphical representation of this moderating effect mechanism.

shows the effect of the predictor (Perception of Quality of care) on the dependent variable treatment compliance at different levels of the moderator (Task-oriented coping)

## Discussion

Our study examined the relationship between coping strategies, perceived quality of care, and treatment adherence among hypertensive patients. The analysis suggests that age, which influences both physical capacity and mental maturity, plays a significant role in treatment adherence. Older individuals tend to exhibit better adherence to treatment regimens compared to younger ones, potentially due to societal expectations of conformity and a heightened sense of maturity. In addition to maturity, an increased awareness of health-related risks may further motivate older individuals to adhere more strictly to prescribed treatments. As they are more likely to experience complications and suffer from various diseases, adherence becomes a more urgent priority. However, despite these plausible explanations, our study did not find statistically significant differences between age, higher blood pressure, and the increased prevalence of chronic conditions, as noted in other studies [23].

Although existing literature suggests that adherence to antihypertensive medications greatly improves blood pressure (BP) control and minimises complications [24,25], this study did not find similar effects. While adherence is important in managing hypertensive patients, it may not directly influence better BP control or reduce comorbid conditions. The non-significant correlations observed in the study may reflect several underlying issues, such as the poor efficacy of the medications, differential patient responses to treatment, or external factors like stress and diet, which may independently impact health outcomes. The relationship between BP and comorbidities, however, was marginally significant (p-value = 0.065), supporting studies that suggest poor hypertension control increases the risk of comorbid cardiovascular and metabolic diseases. Although no strong link was found between admission rates and comorbidities, recent literature, including Das et al., (2020) [26], indicates that comorbid conditions can increase hospitalisation risks. This may suggest

**Table 5. Simple slope analysis.**

|  | Estimate | SE | Z | p |
| --- | --- | --- | --- | --- |
| Average | 0.0913 | 0.02 | 4.57 | <.001 |
| Low (-1SD) | -0.0347 | 0.0284 | -1.22 | 0.222 |
| High (+1SD) | 0.2173 | 0.0262 | 8.29 | <.001 |

that limited access to healthcare or delayed care-seeking behaviours are contributing factors, with patients often avoiding hospital visits until their condition worsens.

Gender differences in self-management and treatment outcomes were evident in this study, as females demonstrated a stronger positive perception of quality care, which was associated with greater adherence to treatment regimens compared to males. This finding underscores the role of gender-specific behaviours in health-seeking and treatment adherence. Research suggests that women generally exhibit better health-seeking behaviours, greater engagement with healthcare providers, and a higher likelihood of following medical advice [27]. Their consistent adherence to treatment may also be influenced by caregiving roles, which reinforce the importance of maintaining personal health. This aligns with previous studies indicating that women are more likely to adhere to treatment regimens for chronic conditions such as hypertension [28]. In contrast, men tend to show lower adherence rates, potentially due to perceptions of invincibility, competing priorities, or societal expectations that may discourage frequent healthcare visits [29]. These factors highlight the need for gender-tailored interventions that address the specific barriers men face in adhering to treatment plans. The differences in the correlation between care quality and adherence among genders also emphasise the importance of healthcare experiences. For females, trust in the quality of care directly enhances adherence, suggesting that patient-centred care improves treatment outcomes [30,31]. The weaker link for males may indicate less engagement with healthcare providers or a perception that treatment is not immediately relevant.

The study further reinforces the essential role of perceived quality of care in determining treatment adherence. When patients perceive care as trustworthy, patient-centred, and involving effective communication, they are more likely to adhere to treatment regimens. Positive perceptions of care foster a sense of trust and commitment to the healthcare process, significantly enhancing adherence, as observed in this study ($\beta = 0.092$, $p < 0.001$). This finding also aligns with previous research emphasising that high-quality care, particularly for chronic conditions like hypertension, has a profound impact on patient engagement and long-term adherence [32].

Among the coping strategies examined, task-orientated coping emerged as the only strategy that positively moderated the relationship between perceived quality of care and treatment adherence. This means that the positive impact of quality care on adherence was strengthened when patients engaged in specific, goal-directed behaviours aligned with their treatment plans. This finding is consistent with the Individual and Family Self-Management Theory (IFSMT), which highlights the role of both internal factors (such as coping mechanisms) and external influences (such as healthcare provider support) in shaping successful self-management [17,18,33]. In contrast, avoidance-orientated coping, which involves denial or disengagement, and emotion-orientated coping, which focuses on venting emotions, did not significantly impact treatment adherence. This suggests that passive coping strategies may be less effective in ensuring long-term adherence, consistent with prior research linking emotion-focused and avoidance coping to negative health outcomes [34].

### Study implications

Taken together, these findings have important implications for hypertension management. The moderating effect suggests that improving coping skills in patients, especially task-oriented strategies, could be a key intervention for enhancing the effectiveness of hypertension treatment programs. Healthcare providers can incorporate coping strategy training into patient care plans, enabling them to better manage stress and engage more actively with treatment. This approach would help bridge the gap between perceived quality of care and actual treatment adherence, ultimately improving health outcomes for patients with hypertension. Enhancing communication and building trust in healthcare settings are also essential for improving treatment adherence. Additionally, demographic factors such as age and gender must be considered when tailoring treatment approaches to ensure they are relevant and effective. Healthcare providers should also integrate coping strategies into care plans, addressing not only hypertension but also any comorbid conditions. However, further research is needed to explore additional factors, such as social support and health literacy, that may influence the

relationship between quality of care and treatment adherence. These insights could lead to more personalised, targeted interventions that better support patients in managing their hypertension.

## Strengths and limitations

This study provides valuable insights into hypertension management, focusing on adherence, gender differences, and coping strategies. The robust statistical methods employed enhance the reliability of the findings, particularly in the context of task-oriented coping and gender-specific interventions, which hold significant practical implications for healthcare practitioners. However, several methodological limitations should be noted, including the use of an unrepresentative sample, reliance on self-report measures. Additionally, the study could not explore the impact of important covariates such as socioeconomic status and health literacy, which are crucial factors in understanding treatment adherence.

## Conclusion

These findings underscore the complexities of hypertension management in the Ghana, highlighting the interplay between treatment adherence, gender disparities, coping strategies, and perceived quality of care. Task-orientated coping emerged as a key factor in improving adherence, whereas other coping strategies had limited impact, reinforcing the need for proactive self-management approaches in chronic disease care. Gender differences further emphasise the importance of tailored interventions, particularly to address lower adherence rates among males. Overall, these results support the need for socially sensitive and personalised strategies to overcome barriers to effective hypertension management.

## Supporting information

**S1 Text. Collinearity diagnostics of regression models.** This file includes variance inflation factors (VIF), tolerance values, and condition indices for all independent variables used in the regression models.
(DOCX)

**S2 Text. Variance proportions among predictor variables.** This file presents the variance decomposition proportions used to assess multicollinearity among the predictors included in the final models.
(DOCX)

## Acknowledgments

We would like to express our sincere gratitude to all participating hospitals for their invaluable support and collaboration in this study. A special thanks to the hypertensive patients who generously shared their time, insights, and experiences. Your participation and dedication were essential to the success of this research.

## Author contributions

**Conceptualization:** Prince Owusu Adoma, Francis Aacquah.

**Formal analysis:** Prince Owusu Adoma, Francis Aacquah, Abubakari Yakubu.

**Investigation:** Francis Aacquah, Christopher Bawiah, Samuel Oke.

**Methodology:** Prince Owusu Adoma, Francis Aacquah, Abubakari Yakubu.

**Project administration:** Prince Owusu Adoma.

**Resources:** Prince Owusu Adoma.

**Software:** Prince Owusu Adoma, Francis Aacquah, Samuel Oke.

**Supervision:** Abubakari Yakubu.

**Validation:** Prince Owusu Adoma, Francis Aacquah, Christopher Bawiah, Samuel Oke.

**Writing – original draft:** Prince Owusu Adoma, Francis Aacquah, Christopher Bawiah, Samuel Oke, Abubakari Yakubu.

**Writing – review & editing:** Prince Owusu Adoma, Abubakari Yakubu.

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
