## [Decision Letter · Decision Letter 0]

17 Jun 2025

PGPH-D-25-01016

Beyond Medical Prescription: Unveiling Coping Strategies' Role Between Perceived Quality of Care and Treatment Adherence among Hypertensive Patients

Dear Dr. Adoma,

Thank you for submitting your manuscript to PLOS Global Public Health. After careful consideration, we feel that it has merit but does not fully meet PLOS Global Public Health’s publication criteria as it currently stands. Therefore, we invite you to submit a revised version of the manuscript that addresses the points raised during the review process.

This is an important study regarding the role of coping strategies in the relationship between perceived quality of care and treatment adherence in hypertensive patientsThere is  conflict in the responses by reviewers, but all the three have raised certain points which you should give attention and responses to make the manuscript better.I would like to congratulate the authors that this is a manuscript helps to shed light on the psychosocial dimensions of chronic disease management taking the example of a very common disease, hypertension

We look forward to receiving your revised manuscript.

Kind regards,

Suma Krishnasastry, MBBS, MD,DNB, FRCP (Edin)

Academic Editor

Journal Requirements:

1. In the online submission form, you indicated that [Data used in this study are available on request.].

a. In a public repository,

b. Within the manuscript itself, or

c. Uploaded as supplementary information.

Additional Editor Comments (if provided):

Reviewers' comments:

Reviewer's Responses to Questions

**Comments to the Author**

1. Does this manuscript meet PLOS Global Public Health’s publication criteria ? Is the manuscript technically sound, and do the data support the conclusions? The manuscript must describe methodologically and ethically rigorous research with conclusions that are appropriately drawn based on the data presented.

Reviewer #1: Yes

Reviewer #2: Yes

Reviewer #3: Partly

2. Has the statistical analysis been performed appropriately and rigorously?

Reviewer #1: Yes

Reviewer #2: Yes

Reviewer #3: Yes

3. Have the authors made all data underlying the findings in their manuscript fully available (please refer to the Data Availability Statement at the start of the manuscript PDF file)?

Reviewer #1: Yes

Reviewer #2: Yes

Reviewer #3: No

4. Is the manuscript presented in an intelligible fashion and written in standard English?

Reviewer #1: Yes

Reviewer #2: Yes

Reviewer #3: Yes

5. Review Comments to the Author

Reviewer #1: A timely study from Ghana on moderating role of coping strategies in the relationship between perceived quality of care and treatment adherence.

1. Suggest to mention the permission arrangement to use the following instruments:

Adherence to Systemic Hypertension (QASH) Treatment

Service Quality (SERVQUAL) Questionnaire

Coping Inventory for Stressful Situation-21 (CISS-21) Questionnaire

2. Include a reference for Service Quality (SERVQUAL) Questionnaire

Reviewer #2: Thank you for the opportunity to review this thoughtful and well-constructed manuscript. The study makes a meaningful contribution to the literature on the psychosocial dimensions of chronic disease management, with a particular focus on hypertension. By examining the role of coping strategies as potential mediators or moderators between perceived quality of care and treatment adherence, the research highlights an important and often overlooked aspect of patient behavior. The topic is both timely and relevant, especially in low-resource settings, and the manuscript addresses it with academic rigor and clarity.

Reviewer #3: 1. The introduction is concise and avoids unnecessary details. However, the authors should kindly clarify what they meant by 'high level of adherence' in the second sentence of the introduction to enhance clarity of the sentence.

2. The authors should specify the 'themes' referenced from 'Bhandari et al. [2]' in line 49, as the sentence lacks coherence. Specifying the themes will ensure clarity and context.

3. The authors indicated in the manuscript that the prevalence of hypertension in Ghana is high, and hypertension care is typically generalised rather than tailored to individual needs. The authors should substantiate these assertions by providing appropriate references.

4. The authors should cite their source for the global NCDs target (reduction of HPT by 33% between 2010 and 2030) for easy reference.

5. The authors should define what they mean by 'coping strategies'.

6. The authors indicated that, on average, 150 to 200 outpatients with various cardiovascular diseases attend the Diabetic and Hypertensive Clinic. They should kindly indicate the timeframe for this estimate of 150 to 200. For example, do these estimated 150 to 200 average CVD cases attend the clinic daily, weekly, or monthly? Clarifying this will provide a better understanding of the sample size estimate for the study and statistical rigor of the findings.

7. The authors should re-evaluate the study design used. They mentioned using a retrospective observational study in the methodology section of the main body of the manuscript. However, their data collection procedure, capturing their response at a point in time, indicate that the study is more accurately characterised as a cross-sectional study rather than the cohort they claimed was adopted. This is despite the fact that the defined population for the study consisted of a cohort of HPT patients.

8. In the statistical analysis subsection of the manuscript, the authors indicated they describe the demographic characteristics of the study population by using measures of central tendency. The authors should kindly specify the measure of central tendency used in describing these demographic characteristics. Specifying the measures used will ensure appropriate interpretation of the data, including their distribution and nature, and will justify the choice of the statistical tests used in the study.

9. The authors should consider specifying the statistical test used for the correlation analysis in the statistical analysis subsection and justify their choice with appropriate references, rather than stating just the visualisation tool (scatter plot). The scatter plot, moreover, depicted only the correlation between the gender differences in perception of quality of care and treatment adherence.

10.The authors should consider describing the demographics with median instead of mean, as the mean may be misleading considering the assumed nature of the data (non-parametric assumption).

11. To enhance the understanding of the statistical tests used in the analysis, the authors should provide a clear definition or description of their variables, including the types of variables and their coding for the analysis. For example, regarding the 'comorbidities' variable, it should be specified whether it was treated as continuous or discrete, coded as categories a, b or c, represented in binary, or optioned yes or no, or quantified as the number of comorbidities, etc.

12. The authors should kindly justify the choice of variable inclusions in the regression models with reference. This justification should be incorporated in the methodology section.

13. The authors indicated in the limitations section that they employed a cross-sectional design, which they noted as a limitation. However, in the methodology section, they stated that they utilised a retrospective observational cohort design. The authors again indicated in their abstract in the cover page that they used a prospective observational cohort study; contrary to what they indicated in the main manuscript. This discrepancy raises concerns about validity of their findings and the methodology used. The authors should rectify this inconsistency to ensure transparency and maintain integrity of their study.

14. The abstract should specify the statistical tests and tools used in the study.

15. What the authors consider as a 'statistically significant p-value should be clearly defined, as they reported a p=0.065 as a significant correlation (BP and comorbidities). This will ensure comprehensive understanding of the findings.

6. PLOS authors have the option to publish the peer review history of their article (what does this mean? ). If published, this will include your full peer review and any attached files.

**Do you want your identity to be public for this peer review?** For information about this choice, including consent withdrawal, please see our Privacy Policy .

Reviewer #1: No

Reviewer #2: No

Reviewer #3: No

---

## [Decision Letter · Decision Letter 1]

9 Sep 2025

Beyond Medical Prescription: Unveiling Coping Strategies' Role Between Perceived Quality of Care and Treatment Adherence among Hypertensive Patients

PGPH-D-25-01016R1

Dear Dr Adoma,

We are pleased to inform you that your manuscript 'Beyond Medical Prescription: Unveiling Coping Strategies' Role Between Perceived Quality of Care and Treatment Adherence among Hypertensive Patients' has been provisionally accepted for publication in PLOS Global Public Health.

Best regards,

Suma Krishnasastry, MBBS, MD,DNB, FRCP (Edin)

Academic Editor

Reviewer #1:

Reviewer #3:

Reviewer Comments (if any, and for reference):

Reviewer's Responses to Questions

**Comments to the Author**

1. If the authors have adequately addressed your comments raised in a previous round of review and you feel that this manuscript is now acceptable for publication, you may indicate that here to bypass the “Comments to the Author” section, enter your conflict of interest statement in the “Confidential to Editor” section, and submit your "Accept" recommendation.

Reviewer #1: (No Response)

Reviewer #3: All comments have been addressed

2. Does this manuscript meet PLOS Global Public Health’s publication criteria ? Is the manuscript technically sound, and do the data support the conclusions? The manuscript must describe methodologically and ethically rigorous research with conclusions that are appropriately drawn based on the data presented.

Reviewer #1: (No Response)

Reviewer #3: Yes

3. Has the statistical analysis been performed appropriately and rigorously?

Reviewer #1: (No Response)

Reviewer #3: Yes

4. Have the authors made all data underlying the findings in their manuscript fully available (please refer to the Data Availability Statement at the start of the manuscript PDF file)?

Reviewer #1: (No Response)

Reviewer #3: Yes

5. Is the manuscript presented in an intelligible fashion and written in standard English?

Reviewer #1: (No Response)

Reviewer #3: Yes

6. Review Comments to the Author

Reviewer #1: Include a reference for Service Quality (SERVQUAL) Questionnaire in the following sentence:

"The questionnaire was developed from three different pre-existing instruments: Adherence to Systemic Hypertension (QASH) Treatment [19] to measure treatment adherence; Service Quality (SERVQUAL) Questionnaire to determine quality of service, and Coping Inventory for Stressful Situation-21 (CISS-21) Questionnaire [20] to measure coping strategies used for hypertension management".

Reviewer #3: Thank you for your comprehensive revisions and for addressing the comments provided in the initial review.

With regard to the abstract, while I maintain that a brief mention of the primary statistical tests and analytical tools employed could enhance transparency and assist readers in quickly assessing the methodological rigor of the study, I understand and respect the authors’ decision to omit these details.

Given that all substantive concerns have been addressed, I am satisfied with the current version of the manuscript. But the authors should quickly check the abstract in the initial table at the beginning of the manuscript containing the information about the manuscript and the authors, as they wrote ‘A hospital-based prospective observational cohort study’ but ‘A hospital-based retrospective observational cohort study’.

7. PLOS authors have the option to publish the peer review history of their article (what does this mean? ). If published, this will include your full peer review and any attached files.

**Do you want your identity to be public for this peer review?** For information about this choice, including consent withdrawal, please see our Privacy Policy .

Reviewer #1: No

Reviewer #3: No
